# Influence of Substrates on the Quality of *Hermetia* Meal for Fish Meal Substitution in Nile Tilapia *Oreochromis niloticus*

**Sven Wuertz** [1,*] **, Cem Hinrich Pahl** [1] **and Werner Kloas** [1,2]

1   Department Fish Biology, Fisheries and Aquaculture, Leibniz-Institute of Freshwater Ecology and Inland Fisheries, Müggelseedamm 310, 12587 Berlin, Germany
2   Faculty of Life Sciences, Humboldt University, Invalidenstr. 42, 10099 Berlin, Germany
*   Correspondence: wuertz@igb-berlin.de

**Abstract:** Commercially produced black soldier flies (*Hermetia illucens*) represent a promising fish meal substitute, particularly in the context of using agricultural by-products and waste. Here, the culture of *Hermetia* maggots on five selected substrates (potato protein (P) as a by-product of starch production, rapeseed oil cake (R) from rape oil production, maize silage (M), soybean (S) meal and, as a control, concentrated chicken feed (C)) were evaluated, assessing the growth performance of *Hermetia* maggots related to the overall production and the nutritional composition of the respective meal. Subsequently, their use as ingredients in aquafeed formulations was evaluated in a feeding trial with juvenile Nile tilapia *Oreochromis niloticus*, assessing the growth performance of the fish. Substrates used for *Hermetia* culture significantly affected the growth and development of the maggots, revealing substantial differences in the meal quality. Still, if incorporated in isonitrogenous and isocaloric diets (33% crude protein, 21–22 MJ/kg) replacing 75% of the fishmeal protein in the formulated diets, no significant differences in growth performance of the fish were observed compared to the fishmeal control. As a conclusion, substrates clearly affect the production yield and the composition of maggots. Nevertheless, this can be compensated by feed formulation as demonstrated by the feeding trial.

**Keywords:** black soldier fly; protein alternative; fish meal substitution; sustainability; feed ingredient; alternative protein; tilapia

## 1. Introduction

Substantial price increases in fish meal (FM) and the limited supply on a global scale reflect the need for alternative feed ingredients, preferably by the use of foodstuff not intended for human consumption [1–4]. In fish compound feed, the inclusion of fishmeal is constantly decreasing due to the limitation of resources. Indeed, in commercial salmon feed, FM decreased from 300 g/kg in 2006 to approximately 150 g/kg in 2012 [5]. In the past, vegetable by-products such as cottonseed oilcake [6], potato protein [3,4,7,8] or rapeseed protein concentrate [1,9] have successfully been used in fishmeal replacement. Still, vegetable ingredients exhibit unfavorable properties such as divergent amino acid profiles, lack of essential metabolites found in animal proteins (e.g., creatine), antinutritive secondary metabolites or low digestibility [10]. Insect meals are protein rich, have a good amino acid profile and provide trace elements [11,12]. Furthermore, insect meals have a low environmental impact [13,14]. Industrial scale farms produce fewer greenhouse gases, have lower ammonia emissions, have a lower water footprint and need less space than classical livestock for a respective yield [13,15]. Among the few species cultured so far [15,16], the Black soldier fly *Hermetia illucens* (BSF) is ideally suited as a feed ingredient. First, it can be grown on agricultural waste streams, converting a variety of low valued substrates into high quality protein biomass [17–19]. This added benefit of waste recycling may thereby solve two problems at once, the need for good quality feed ingredients and the growing

concern of costly food waste management. Secondly, the production process can be widely automated [20,21].

BSF larvae have the potential for large-scale utilization in feed due to their excellent nutrient profile (42–58% crude protein, 33–39% crude lipids; [5,11,12,22,23]). Recently, BSF meal (BSFM) has been studied in a variety of livestock including chicken [24,25], pigs [26,27] and beef [28]. In fish, BSFM as an ingredient in feed has been assessed in African catfish *Clarias gariepinus* [29,30], barramundi *Lates calcarifer* [31,32], rainbow trout *Oncorhynchus mykiss* [33–35], salmon *Salmo salar* [36], sea bream *Sparus aurata* [35], Striped catfish *Pangasianodon hypophthalmus* [37], tench *Tinca tinca* [35], turbot *Scophthalmus maximus* [11] and Nile tilapia *Oreochromis niloticus* [38]. In tilapia, BSFM replaced up to 54% of the protein without significant differences compared to a soybean meal control at 42% crude protein (CP) [38].

Tilapia is the third most farmed fish species worldwide with a production of approximately 4.5 million tons annually [39]. It has a good feed conversion in routine farming [40,41], fast growth [42–44], good acceptance of alternative protein sources [6,42,45], high stress resistance [46–48], tolerance towards suboptimal water quality [49–52] and relatively low susceptibility to diseases [42]. Taken as a whole, tilapia is a robust fish species with excellent potential in global aquaculture [53]. As an omnivorous species, tilapia has a protein requirement of approximately 35–45% in juveniles decreasing to 20–30% in subadults [54–56]. Nevertheless, for optimal performance of the broodstock, 35–45% protein is required [57–60]. Although lipid requirements highly depend on lipid source and dietary protein, 10–15% lipids have been recommended for maximal growth performance. Nevertheless, farmers often use lower lipid contents of 6–8% [54].

In the present study, we studied the influence of the substrate on the yield and quality of the BSFM harvested, and assessed the potential impact of the respective BSFM in a classical feeding trial with tilapia.

## 2. Materials and Methods

### 2.1. BSF Reproduction

The BSF were obtained from a commercial company (Hermetia Futtermittel GbR, Baruth/Mark, Germany). For the production of sufficient fertile adults and clutches of eggs, a large-scale culture system was used: First, maggots were grown in plastic dishes (24.6 cm length × 18.1 cm width × 5.0 cm height) at 26 °C and humidity >60%. For the pupation, prepupae were collected and transferred to a new dish (38.5 cm × 27.5 cm × 14.0 cm) filled with coconut humus and incubated at 26 °C. After 14 d, adult flies were transferred into a flight container (235 cm × 110 cm × 275 cm) exposed to direct sunlight (4 m$^2$ window), because sunlight is required for successful mating [61,62]. In addition, as recommended by Zhang et al. [62], six 58 W halogen lights were installed. Illumination was provided between 9:00 and 18:00. Humidity (>60%) was adjusted with a humidifier Luxven CF-2728-B. Furthermore, two funnels were fitted with foam and constantly wetted with a water pump Eheim 2217 (600 L/h). Females oviposited in flutes of strips of corrugated cardboard, which were attached to the wall of a bucket containing wet chicken feed as an attractant as suggested by Tomberlin and Sheppard [61].

### 2.2. Substrate Utilization

Selected substrates were obtained as follows: raw potato protein (Kartoffelprotein K5, Emsland-Stärke GmbH, Emlichheim), chicken feed (Legehennenmehl, Krausland, Ebertsheim), maize silage (Milchhof Mendler, Berlin), rapeseed cake (Ölmühle Hans-Hermann Wagner, Straßkirchen-Schambach), soybean meal (Scharnebecker Mühle DiHa GmbH, Scharnebeck). Raw potato protein contains relatively high concentrations of glycoalkaloids. Chicken feed has been used in several studies [17,18,63] as an ideal substrate and is used here as a reference. Soybean meal is a high-quality plant protein source that is frequently used in aquafeeds and is considered as ideal here.

For the determination of growth, two egg clutches per substrate were transferred to a container (27.5 cm × 18.5 cm × 14.0 cm) filled with the respective substrate. Humidity of the substrate was controlled every 24 h and adjusted if necessary. Every 4 d over a period of 28 d (which allows the larvae to reach the prepupal stage), ten larvae were randomly picked with spring steel tweezers, cleaned and weighed to the nearest 0.1 mg with an analytical balance Sartorius MC 210 S (detection limit = 0.01 mg).

### 2.3. Mortality Test

For each substrate, mortality was determined in triplicate. Therefore, 100 newly hatched larvae (approximately 1 day old) were transferred with a fine brush to Petri dishes filled with 100 g of the respective substrate. Humidity was established by regular spraying three times a day. Cumulated mortalities over 28 d were recorded.

### 2.4. Tilapia Feeding Trial

For the production of BSFM, as many egg clutches as possible were transferred to a container (55.8 cm × 36.0 cm × 19.8 cm) at 26 °C, containing the respective substrate in excess. Metabolization of substrate was comparable and substrate was exchanged every second day. All substrates were humified sufficiently after weighing. Prepupae were continuously harvested, washed for 15 min with a mesh, subsequently allowed to dry at room temperature for 60 min, then weighed and frozen at −80 °C. We estimated the required substrate per kg larvae harvested (Table 1). For the production of the experimental diets, larvae were lyophilized for 120 h with a Zirbus Technology Sublimator 3 × 4 × 5. Dried larvae were homogenized with a vibrating cup mill (Fritsch Pulverisette 9) for 2 × 30 s and stored in an airtight plastic box at −20 °C.

**Table 1.** Approximate quantity of substrate required for the production of 1 kg BSF (wet weight).

|   | Substrate | Quantity of Substrate [kg/kg BSFM] |
|---|---|---|
| C | chicken feed | 1.980 |
| P | potato protein | 3.830 |
| M | maize silage | 3.190 |
| R | rapeseed oil cake | 2.970 |
| S | soybean meal | 1.850 |

Proximate composition of the ingredients was determined after freeze drying for 120 h. Analysis for dry matter (DM), ash, crude protein (CP), crude fat (CF) and crude fiber was performed according to the EU guideline EC152/2009. In brief, CP was determined by the Dymar method with an elemental analyzer Vario Max TNS. For all ingredients and the experimental feeds, N × 6.25 was used, except for the wheat ingredients (N × 5.7). CF was determined after hydrolysis with hydrochloric acid followed by petrol ether extraction with a Soxleth extraction system (Soxtec System HAT 1043 Extraction Unit + Soxtec HAT 1046 Service Unit). DM was quantified with an analytical balance (Sartorius MC 210 S d = 0.01 mg) after drying at 105 °C (WTC Binder 7200) until a constant weigh was observed. Ash was determined after 12 h incineration at 750 °C (Rapid Incinerator SVD 95, 2500 W). Gross energy was quantified in bomb calorimeter Parr 6400.

For the feeding trial, six—one fishmeal control diet (FM) and five BSFM-based diets (BC, BP, BM, BR, BS)—isocaloric and isonitrogenous diets were formulated (Table 2). All diet ingredients were mixed and experimental diets were pressed to pellets of 1 mm in diameter using a pellet-press (Alexanderwerke, SKM, Remscheid, Germany).

**Table 2.** Ingredients (g/kg) and gross composition (g/kg DM) of the fishmeal control and the five experimental diets. BSFM—Black soldier fly meal, NfE—nitrogen free extract (100 − (crude protein + crude fat + crude ash)).

| *Experimental Groups:* | FM (Control) | BM | BC | BR | BP | BS |
|---|---|---|---|---|---|---|
| Protein percentage BSFM [%] | 0 | 75 | 75 | 75 | 75 | 75 |
| Protein percentage fish meal [%] | 95 | 20 | 20 | 20 | 20 | 20 |
| *ingredients:* | | | | | | |
| wheat gluten [1] [g/kg] | 20.00 | 20.00 | 20.00 | 20.00 | 20.00 | 20.00 |
| wheat starch [1] [g/kg] | 328.96 | 240.42 | 166.07 | 283.82 | 297.72 | 273.3 |
| fish meal [2] [g/kg] | 476.56 | 95.31 | 95.31 | 95.31 | 95.91 | 95.91 |
| BSFM maize silage [g/kg] | 0 | 639.77 | 0 | 0 | 0 | 0 |
| BSFM chicken feed [g/kg] | 0 | 0 | 714.12 | 0 | 0 | 0 |
| BSFM rapeseed oil cake [g/kg] | 0 | 0 | 0 | 576.37 | 0 | 0 |
| BSFM potato protein [g/kg] | 0 | 0 | 0 | 0 | 462.47 | 0 |
| BSFM soybean meal [g/kg] | 0 | 0 | 0 | 0 | 0 | 466.89 |
| sunflower oil [3] [g/kg] | 170.00 | 0.00 | 0.00 | 20.00 | 120.00 | 140.00 |
| vitamine mix [4] [g/kg] | 2.50 | 2.50 | 2.50 | 2.50 | 2.50 | 2.50 |
| mineral mix [4] [g/kg] | 2.00 | 2.00 | 2.00 | 2.00 | 2.00 | 2.00 |
| *Proximate composition* | | | | | | |
| crude protein | 330.98 | 330.85 | 330.50 | 330.89 | 330.72 | 330.82 |
| crude fat | 220.15 | 211.56 | 235.13 | 215.35 | 215.35 | 220.18 |
| crude ash | 85.99 | 62.34 | 120.67 | 60.56 | 60.56 | 58.15 |
| NfE | 362.88 | 395.25 | 313.70 | 393.20 | 393.20 | 390.85 |
| Gross energy [MJ/kg] | 21.4 | 21.70 | 20.94 | 22.02 | 22.02 | 22.04 |

[1] Minerva Handelsgesellschaft mbH, Reinhard Luginger, Holzlandstraße 1.7, 04779 Calbitz. [2] VF Cuxhaven, Germany. [3] Associated Oil Packers GmbH, Klötzerstraße 28–32, 01587 Riesa. [4] AllerAqua, Christiansfeld, Denmark.

For the feeding trial, 126 juvenile tilapia (13.22 g ± 2.23 g) were randomly stocked to eighteen 37.5 L tanks, arranged as two recirculation systems with a foam filter and a biofilter with clay granules (10% water exchange, 2.5 V water turnover). Each treatment was assessed in triplicate. Temperature was controlled with a 600 W heater ($T_{RAS1}$ = 24.8 ± 0.2 °C, $T_{RAS2}$ = 24.9 ± 0.2 °C). Fish were fed at 2.5% of their body weight once a day. To adjust the feed ratio, fish biomass was determined every week. $O_2$ (>90%), pH (7.1–7.4) and T were monitored daily with a HQ40d multimeter (Hach Lange); $NO_2^{-}$-N by the diazotization method and total ammonia nitrogen (TAN) by the salicylate method every three days using the respective pillow kits (Hach Lange). The photoperiod was set to 14 h light: 10 h dark.

*2.5. Data Analysis*

All data are presented as mean ± standard deviation (SD) of n replicates. Data were analyzed for normal distribution by D'Agostino and Pearson omnibus normality test and equal variance by Kruskal-Wallis one way analysis of variance. Multiple comparison was carried out by parametric Tukey's test or non-parametric Dunn's Test. All statistical tests considered $p < 0.05$ and were performed using GraphPad Prism 4.03 (GraphPad Software Inc., San Diego, CA, USA).

**3. Results**

Mating and oviposition were observed continuously and after the second generation of flies, sufficient egg clutches could be collected. Hatching occurred after 4–5 d with a weight between 0.02 and 0.04 mg (Figure 1).

Dependent on the substrate (Table 3), mortalities were 14% in S, 18% in C, 26% in R and 32% in M. Highest mortalities were observed in larvae grown on potato protein with 47% mortality. Correlated with the mortalities, growth performance after 28 d was best in C and S groups with 176 ± 34 mg and 119 ± 11 mg (Figure 2), respectively. Lowest weight was observed in P with 16 ± 10 mg, followed by M with 69 ± 24 mg and R with 90 ± 37 mg. Specific growth rate ranged between 20.8 ± 3.3% d⁻¹ in P, 26.4 ± 2.1% d⁻¹

in M, 26.8 ± 2.0% d$^{-1}$ in R, 28.5 ± 1.4% d$^{-1}$ in S and 29.9 ± 1.5% d$^{-1}$ in C. Taken as a whole, best performance was observed in C, followed by S. Lowest growth performance was observed in P, which also revealed the highest mortality.

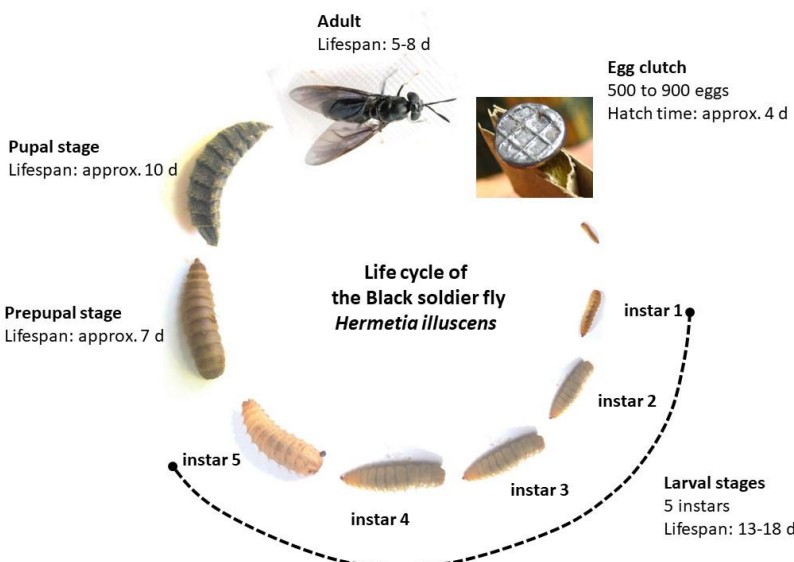

**Figure 1.** Life cycle of the Black soldier fly *Hermetia illuscens*, comprising the egg stage, five larval stages (instar 1–5), the prepupal stage, the pupal stage and the imago, modified from de Smets et al. [64] and Lievens et al. [65].

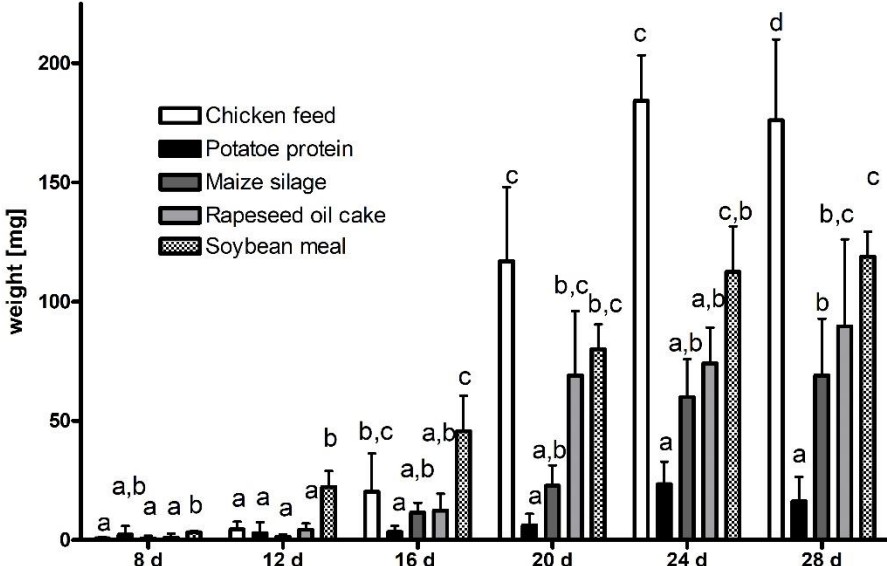

**Figure 2.** Weight (mean ± SD, *n* = 10) of Black soldier fly larvae *Hermetia illuscens* feeding on chicken feed (control), potato protein (P), maize silage (M), rapeseed oil cake (R), soybean meal (S) substrate over an experimental period of 28 d. Significant differences between groups at a given timepoint are indicated by different letters (*p* > 0.05, *n* = 10).

Highest CP (Table 4) was observed in the meal produced on potato protein (54.35 ± 0.05%) and soybean meal (53.84 ± 0.03%). Meal produced on chicken feed revealed the lowest protein content at 35.20 ± 0.03%, followed by maize silage (39.29 ± 0.02%) and rapeseed oil cake (43.61 ± 0.02%). DM ranged between 31.12 ± 1.6% in S and 38.66 ± 1.9% in rapeseed oil cake. Crude fiber was highest in maize silage with 32.24 ± 0.11% and lowest in soybean meal with 14.94 ± 0.23%.



**Table 3.** Growth performance and mortalities of Black soldier fly larvae *Hermetia illuscens* feeding on chicken feed (control, C), potato protein (P), maize silage (M), rapeseed oil cake (R), soybean meal (S) substrate over an experimental period of 28 d. Significant differences between groups are indicated by different letters ($p > 0.05$, $n = 10$).

| Group: | C | P | M | R | S |
|---|---|---|---|---|---|
| $W_i$ [1] | $0.04 \pm 0.02$ | $0.04 \pm 0.02$ | $0.04 \pm 0.02$ | $0.04 \pm 0.02$ | $0.04 \pm 0.02$ |
| $W_f$ [2] | $176.1 \pm 34.0$ [c] | $16.2 \pm 10.3$ [a] | $69.0 \pm 23.8$ [b] | $89.6 \pm 36.5$ [b] | $118.8 \pm 10.6$ [b,c] |
| WG [3] | $176.0 \pm 34.0$ [c] | $16.1 \pm 10.3$ [a] | $69.0 \pm 23.8$ [b] | $89.6 \pm 36.5$ [b] | $118.8 \pm 10.6$ [b,c] |
| SGR [4] | $29.9$ [a] $\pm 1.5$ [c] | $20.8 \pm 3.3$ [a] | $26.4 \pm 2.1$ [a,b] | $26.8 \pm 2.0$ [b] | $28.5 \pm 1.4$ [b,c] |
| M [5] | 18% [c,d] | 47% [a] | 26% [b,c] | 32% [a,b] | 14% [d] |

[1] Initial weight [mg]. [2] Final weight [mg]. [3] Weight gain [mg] = $[W_f - W_i]$. [4] Specific growth rate (% d$^{-1}$) = $100 \times (\ln(W_f) - \ln(W_i))/$feeding days. [5] Mortality (%) = $(100 - $ (number of larvae on 28 d))$/100)$.

**Table 4.** Composition of the Black soldier fly meal produced from flies grown on chicken feed (control, C), potato protein (P), maize silage (M), rapeseed oil cake (R), soybean meal (S) substrate over an experimental period of 28 d. DM-dry matter [%], CP—crude protein [%], CF—crude fat [%], A—ashes [%], NfE—nitrogen free extract [$100 - (CP + CF + A)$]. Significant differences between groups are indicated by different letters ($p > 0.05$, $n = 3$).

| Group | DM | CP | CF | A | NfE | Gross Energy [MJ/kg] |
|---|---|---|---|---|---|---|
| C | $34.8 \pm 1.7$ [a,b] | $35.2 \pm 0.03$ [a] | $31.5 \pm 0.4$ [c] | $14.4 \pm 0.8$ [c] | $18.9 \pm 1.2$ [a] | $22.6 \pm 0.3$ [a,b] |
| P | $32.5 \pm 1.6$ [a] | $54.4 \pm 0.05$ [d] | $18.4 \pm 0.1$ [b] | $8.0 \pm 0.6$ [a,b] | $19.3 \pm 0.8$ [a] | $23.0 \pm 0.1$ [b] |
| M | $37.4 \pm 1.9$ [b] | $39.3 \pm 0.02$ [b] | $32.2 \pm 0.1$ [c] | $6.9 \pm 0.1$ [a] | $21.6 \pm 0.2$ [b] | $24.6 \pm 0.2$ [c] |
| R | $38.7 \pm 1.9$ [b] | $43.6 \pm 0.02$ [c] | $31.4 \pm 0.6$ [c] | $7.4 \pm 0.2$ [a] | $17.6 \pm 0.9$ [a] | $25.4 \pm 0.1$ [c] |
| S | $31.1 \pm 1.6$ [a] | $53.8 \pm 0.03$ [d] | $14.9 \pm 0.2$ [a] | $8.6 \pm 0.01$ [b] | $22.7 \pm 0.3$ [b] | $21.7 \pm 0.3$ [a] |

After the feeding trial (Table 5), juvenile tilapia revealed only minor differences in feed conversion ratio (FCR), weight gain (WG), specific growth rate (SGR), protein efficiency ratio (PER) and Fulton's condition factor (FCF). There were no significant differences between groups.

**Table 5.** Growth performance and feed conversion of tilapia *Oreochromis niloticus* fed a fishmeal control diet (FM) and five experimental diets with Black soldier fly (*Hermetia illuscens*) meal produced on different substrates (Black soldier fly meal produced on chicken feed (BC), on potato protein (BP), on maize silage (BM), on rapeseed oil cake (BR), on soybean meal (BS)). There were no statistical differences.

| | FM | BP | BC | BM | BR | BS |
|---|---|---|---|---|---|---|
| $W_i$ [1] | $13.3 \pm 2.1$ | $13.2 \pm 2.3$ | $13.1 \pm 2.2$ | $13.8 \pm 2.6$ | $13.4 \pm 2.1$ | $12.5 \pm 2.1$ |
| $W_f$ [2] | $19.3 \pm 3.2$ | $20.0 \pm 3.1$ | $17.9 \pm 3.0$ | $19.6 \pm 4.8$ | $19.1 \pm 7.3$ | $19.6 \pm 4.0$ |
| SGR [3] | $1.3 \pm 0.02$ | $1.5 \pm 0.3$ | $1.0 \pm 0.06$ | $1.2 \pm 0.2$ | $1.3 \pm 0.08$ | $1.6 \pm 0.3$ |
| FCR [4] | $1.8 \pm 0.03$ | $1.7 \pm 0.4$ | $2.2 \pm 0.1$ | $2.0 \pm 0.3$ | $1.9 \pm 0.1$ | $1.5 \pm 0.3$ |
| PER [5] | $1.7 \pm 0.03$ | $1.9 \pm 0.4$ | $1.4 \pm 0.8$ | $1.5 \pm 0.3$ | $1.6 \pm 0.1$ | $2.0 \pm 0.4$ |
| FCF [6] | $1.8 \pm 0.03$ | $1.8 \pm 0.05$ | $1.8 \pm 0.04$ | $1.8 \pm 0.06$ | $1.9 \pm 0.01$ | $1.8 \pm 0.06$ |

[1] Initial weight (g). [2] Final weight (g). [3] Specific growth rate [% d$^{-1}$] = $100 \times (\ln(W_f) - \ln(W_i))/$feeding days. [4] Food conversion ratio = feed intake (g)/weight gain (g). [5] Protein efficiency ratio = weight gain (g)/Protein intake (g). [6] Fulton's condition factor = $[100 \times W_f/\text{length}^3]$.

## 4. Discussion

After being approved as feeds by the European Union regulation (Commission Regulation (EU) 2017/893 on 24 May 2017) there has been considerable interest in insects such as BSF, mealworms or houseflies. Moreover, both consumer perception and acceptance are positive, due to the natural relationship of the fish as a predator and an insect as prey in the trophic chain [66]. The BSF can be used to convert waste food and agricultural by-products

into a protein-rich ingredient for aquafeed [33,36]. Here, we performed an experiment to study the use of five agricultural by-products as substrates for the BSF larvae. After 28 d of feeding the larvae revealed substantial differences in growth performance, survival, substrate utilization and composition of the meal subsequently produced. The highest mortality (47%) and the lowest growth performance was observed with the P, whereas S and C revealed the best growth performance, lowest substrate utilization and highest survival. Additionally, the composition of the meal was substantially different. Indeed, BSF grown on P had the highest protein content, similar to larvae grown on S, whereas lowest protein content was observed in the larvae grown on C. Consequently, yield and quality of the BSFM highly depends on the substrate. It has been suggested that, during suboptimal food supply, larvae will feed until they have reached the minimum energy reserve required to perform pupal development. This critical weight is defined as the minimal weight at which further feeding and growth are not required for a normal metamorphosis and pupation [67]. The low growth performance observed in the P group is most probably caused by secondary plant metabolites, namely glycoalkaloids. These glycoalkaloids (solanine, chaconine) are natural insect repellents, particularly produced by sunlight exposed potato tissue [68,69]. They also affect fish as antinutritional factors [4]. It is interesting that reduced growth of larvae but not of the respective fish group was observed here, suggesting that the glycoalkaloids do not bioaccumulate in the larvae. One may even speculate that these are metabolized or not incorporated. Nevertheless, protein content of the substrate may also contribute to the low growth performance of larvae. Nguyen et al. [70] have shown that larvae do not develop well on protein-rich substrates such as meat. Still, in our study, larvae fed on protein-rich soybean meal revealed an excellent growth performance.

The protein content observed in larvae feeding on P and S were relatively high compared to the 42–44% reported in previous studies [71–73]. The protein content of larvae feeding on rapeseed oil cake and maize silage are in the range reported by Diener et al. [17,74]. To our knowledge, this is the first experimental proof demonstrating protein contents >50%, dependent on the substrate used. Additionally, to produce meals with high protein levels, protein rich substrates are needed. Nevertheless, all these differences in meal composition can be compensated for during feed formulation for fish.

In the present study, we used entire pupae for the production of the BSFM. As a consequence, our feed has relatively high CF content ranging between 21.2 and 22.0%. This is above the recommended value of 10–15% recommended by El-Sayed [54] for tilapia. Nevertheless, commercial diets often have CF contents ≥ 20% and similar gross energy, for example the tilapia diet Aller Performa (Emsland-Aller Aqua GmbH, Golßen). Due to the high fat content of the BSF, several studies used partially defatted meal [11,38,75]. Recently, Abu Bakar et al. [76] replaced fat sources by BSF oil to up to 100% and observed best growth performance and feed conversion at 25%. Congruently, it has been shown that BSF oil is a valuable replacement for fish oil, particularly if larvae are grown on substrates rich in PUFA such as fish offal [77]. Indeed, the authors even reported an immune stimulation in the BSF oil groups.

As demonstrated in the present study, one may reduce the fat content and enrich the protein content of BSFM by feeding substrates with higher protein contents such as potato protein or soybean meal. Obviously, such a strategy may affect sustainability when substrates are used that are suited for human consumption or already used in feed formulations such as soybean meal. Still, there are a lot of by-products that may be used such as cotton-seed oil cake, blood meal, slaughter waste, inedible plants (e.g., brown algae or microalgae) or feather meal [6,75,77–81]. Depending on the countries, climates and seasons, different substrates can be used. Indeed, BSF seem to metabolize antinutritional metabolites such as chaconine or solanine. Additionally, the biosecurity of insect meal seems to be better than potential ingredients such as slaughter waste. Undoubtedly, questions remain relating to allergenicity and biosecurity of insect meals. Furthermore, as demonstrated, to assure a constant quality of the respective meal the use of a defined, constant substrate is of utmost importance.

It has been shown that total replacement of fish meal reduced protein digestibility and activity of several digestive enzymes in tilapia [82]. In an experiment with low BSFM substitution (max. 80 g/kg), FCR between 2.0 and 2.2, PER between 1.2–1.3 and SGR > 3.1 were reported [83]. In the present study, FCR was better in the BS group (1.5) and comparable in the BC group (2.2). PER was higher at 1.4 in the BC group and 2 in the BS group. In the present study, SGR was relatively low (1.0–1.6%/d). Still, fish in the present study were bigger, 13 g compared to 5–6 g in the previous study. A similar study [38] with defatted BSFM resulted in comparable growth performance and those SGR > 3%/d are rather exceptionally high.

## 5. Conclusions

The influence of the substrate is often underestimated and is here reflected by substantial differences in composition as well as larval yield. Therefore, substrate used in an operation destined to feed formulation should be based on a constant supply of a comparable substrate composition. Thereby, batch to batch variations of BSFM can be minimized. In turn, this reduces the required labor and material costs for the analysis of variable ingredients during feed formulation.

**Author Contributions:** Conceptualization, S.W. and W.K.; methodology, S.W. and C.H.P.; validation, S.W.; formal analysis, C.H.P.; investigation, C.H.P.; resources, S.W. and W.K.; data curation, S.W. and C.H.P.; writing—original draft preparation, S.W.; writing—review and editing, S.W., W.K. and C.H.P.; visualization, S.W.; supervision, S.W. and W.K.; project administration, S.W.; funding acquisition, S.W. All authors have read and agreed to the published version of the manuscript.

**Funding:** This research received no external funding.

**Data Availability Statement:** Data will be made available on request.

**Acknowledgments:** We thank Eva Kreuz for her support during sampling and lab analysis.

**Conflicts of Interest:** The authors declare no conflict of interest.

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
