# Peer review of "Influence of Substrates on the Quality of Hermetia Meal for Fish Meal Substitution in Nile Tilapia Oreochromis niloticus"

_water, doi:10.3390/w14192953_

Round 1

Reviewer 1 Report

The authors present an experiment on the substitution of fish meal with Hermetia meal (Black soldier fly), where the flies were cultured on different substrates. The manuscript is interesting and should be published. Unfortunately, there was no significance in the growth parameters of the fish, here I would check again statistically especially between BC and BS. Furthermore, it would be advisable to add the water parameters of the two recirculation systems in the results to show that the fish were kept under optimal conditions. However, this is not mandatory and should be decided by the authors themselves.  

For further corrections please read the comments in the pdf.

Author Response

We greatly acknowledge the time and effort of the reviewers as well as the editor. This greatly helped us to revise and improve the manuscript. We revised the manuscript as suggested. All changes in tracked changes mode can be found in the attached file of the manuscript. Our response and the line numbers below refer to this tracked changes version of our manuscript.

Reviewer: 1
The authors present an experiment on the substitution of fish meal with Hermetia meal (Black soldier fly), where the flies were cultured on different substrates. The manuscript is interesting and should be published. Unfortunately, there was no significance in the growth parameters of the fish, here I would check again statistically especially between BC and BS. Furthermore, it would be advisable to add the water parameters of the two recirculation systems in the results to show that the fish were kept under optimal conditions. However, this is not mandatory and should be decided by the authors themselves. 

For further corrections please read the comments in the pdf.

We did statistics as described but could not detect a significant difference between the groups in our feeding trial with tilapia. Obviously, we performed a non-parametric multiple comparison test as required.

Water parameters were included in the text: oxygen was >90% and pH ranged between 7.1 and 7.4, which is regarded “ideal” for the experiment. Temperature did not vary between the two RAS:

(TRAS1 = 24.8 ± 0.2 °C, TRAS2 = 24.9 ± 0.2 °C). Fish were fed at 2.5% of their body weight once a day. To adjust the feed ratio, fish biomass was determined every week. O2 (> 90%), pH (7.1-7.4) and T were monitored daily with a HQ40d multimeter (Hach Lange);

All other comments provided were corrected in the track changes version

Reviewer 2 Report

please see attched word. there are MAJOR text editing errors, and a few points to be clarified 

Author Response

We greatly acknowledge the time and effort of the reviewers as well as the editor. This greatly helped us to revise and improve the manuscript. We revised the manuscript as suggested. All changes in tracked changes mode can be found in the attached file of the manuscript. Our response and the line numbers below refer to this tracked changes version of our manuscript.

Reviewer: 2

In general, the paper is interesting. But There are MAJOR TYPING ERRORS : lack of tables, lack of

legend, text missing? 

I regret the use of substrates that ARE used for human consumption or direct use, and not substrates

of lesser value, as you very wisely recommend the use. In the same line, you choose dry substrates,

when the interest of Hermetia is the possible use of wet substrates.  You might want to add a word

explaining the choice of these substrates as a model to gain knowledge and help further work in

different countries with different substrates of low value and changing with season. 

Indeed, we used one substrate that is used for human consumption, soybean meal. Raw potatoe protein is rich in glycoalkaloids and is thus not used for human consumption. Chicken feed was used as a reference since it has been used in several studies with Hermetia. Rapeseed oil cake is a by-product of oil extraction and also not used for human consumption same as maize silage. The following text was added to the manuscript:

Raw potato protein contains relatively high concentrations of glycoalkaloids. Chicken feed has been used in several studies [17, 18, 72] as an ideal substrate and is used here as a reference. Soy bean meal is a high quality plant protein source that is frequently used in aquafeeds and is considered as ideal here.

As well, you have 2 animals, substrates, ingredients and feed. Sometimes the wordings are confusing. 

We believe we established a comprehensive system. Substrates are always referred to as P, M, R, S and C,

Hermetia feeding groups are accordingly named after the substrate grown on P, M, R, S, C. The fish are named differently as FM (fishmeal control), BP, BM, BR, BS, BC as they received Black soldier fly meal produced on the respective substrate. We applied this concept and described it in the text. We believe it is a good compromise to facilitate the reading (due to the uniformly used substrate name) and the unmistakable differentiation between the two animals in our study. In the ingredients table, the respective meal was called BSFM and the respective substrate, avoiding further abbreviations, e.g. BSFM soybean meal.

Abstract : 

  1. why chicken feed as a control ? please detail that point in M&M

Chicken feed has been used in several studies [17, 18, 72] as an ideal substrate and is used here as a reference. This sentence was added to the text.

  1. Subsequentely, the utilization in aquafeeds; Please prefer: Subsequently, their use as

ingredients / Subsequently, their use in aquafeeds formulation

Has been changed

Keywords: you might want to add tilapia

added

Introduction: 

  1. in fish meal (FM) and limited

done

  1. the need for good quality feed ingredients

 done

  1. materiel and methods
  2. BSF Reproduction

done

  1. Why 28 days, please explain

We have chosen 28 d as it allows the pupae to reach the prepupal stage, the stage BSF is harvested. This has been added to the text.

  1. the description of mortality tets is unclear: is it same test as nutrition as results let

understand, or separate tests as theis M1M let understand

This is a separate test with 100 indiviuals in separate boxes as describeeed in detail

  1. if separate tests, add humidity control

Added: Humidity was established by manual spraying three times a day

  1. title better be substrate trial. Please change

We prefer feeding trial because that is exactly what it is.

  1. please add that substrate is in excess

done

  1. Write day in full letters and not just d.

d is the international abbreviation. In line 107 we changed to (1 day old)

  1. Table 1. Why is the substrate in g/kg and not in kg/kg

changed

  Would be wise to have the composition of the differente substrates to help understand the

discussion at the end  121. Proximate composition of the BSFM obtained (ingredients is confusing at this stage of the

reading) 

We do not agree. Tab 4 provides a better clarity.

  1. add a title Tilapia feeding trial

done

  1. parenthesis missing before g/kg. in the Table 2, at the end of line sunflower oil, there is a bar

that should be removed. NfE: what is it? 

Done

Added: NfE – nitrogen free extract [100-(crude protein + crude fat + crude ash)]

  1. results
  2. strange to start with Table 3 results, when you have not addressed Fig 2 results. Please

reorder 

done

  1. space before Correlated to remove

done

  1. Space after weight to remove

done

  1. please specify that this is results of Table 4.

done

  1. missing legend with abcdef

Abbreviations were added to the legend

  1. After the feeding trial on tilapia (Tab.5).

After the feeding trial (Tab. 5), juvenile tilapia revealed only minor differences in feed conversion ratio (FCR), weight gain (WG), specific growth rate (SGR), protein efficiency ratio (PER) and Fulton’s condition factor (FCF).

  1. Title of Table 5 MISSING

Has been added

  1. ?????????????????????????????????

This sentence was deleted

  1. Discussion

Don’t they feed on the dead? Could that explain the high protein content with potato 

You might want to add that protein quality of the substrate influe little on BSFM protein quality 

That is actually not true, high protein substrate result in high protein meal

  1. Moreover, … this sentence os difficult to understand. Please ask around you and change.

Who is their In line 209?  

Moreover, both, consumer perception and acceptance are positive, due to the natural relationship of the fish as a predator and an insect as a prey in the trophic chain [63].

212-219.   This is more Results than Discussion 

It provides a summary of the results which is important for the subsequent discussion

  1. recommended by xx for tilapia. Please add tilapia, as again, it is easy to get lost between BSF

and tilapia 

done

  1. protein content of BSFM. Please add

done

  1. you might want to add here that depending on countries, climates and seasons, different

substrates can be used. 

done

  1. homogeneous is not the best word. Prefer regular or constant in time.

Therefore, substrate used in an operation destined to feed formulation should be based on a constant supply of a comparable substrate composition.

  1. batch to batch is not the best. Refer seasonal, or variations due to substrate change over

time. If I’m correct, it is not minimizing, but correcting via the formulation the variations of BSFM

grown on changing substrates.

Batch to batch variations is exactly the right word. We did not change this.

277 again.   That sentence MUST be better formulated. An economical approach is interesting,

but this sentence is not understandable. I see you want to have a touch of economical comment. But

I don’t understand what you mean. My point of view is that you prove that it is possible to adjust the variations of substrate with a good formulation. So speaking about reducing labour during

formulation is contradictory. Please reformulate as I really appreciate an effort in economical

coment. 

Changed: Thereby, batch to batch variations of BSFM can be minimized. In turn, this reduces the required labor and material costs for the analysis of variable ingredients during feed formulation.

Reviewer 3 Report

The manuscript entitled “Influence of substrates on the quality of Hermetia meal for fish 2 meal substitution in Nile tilapia Oreochromis niloticus” present an interesting study but it cannot be accepted in the present form. A thorough review of the work is necessary to make it publishable in such a prestigious journal. The major criticality is linked to the experimental design. In addition to the tested substrates, it is necessary to use a standard substrate for HI breeding to obtain a control group to compare the obtained results. Generally, the control substrate is the one used by the company that provides HI. The lack of a control group makes the work unacceptable.

Other suggestions

I suggest adding in the introduction the part concerning the description of substrates that have been tested in the literature for the growth of HI, on the basis of which probably the authors have chosen the substrates to be tested in this research.

Pay attention to the significant figures. The general rule is that experimental uncertainty (Standard Deviation) should be rounded to one significant digit. In Tables and along the text, authors reported too many digits. For example, in Table 3 replace “176.07 ± 34.0” with “176 ± 34”; “20.75 ± 3.3” with “20.8 ± 3”, “89.64 ± 36.5” with “90 ± 36” and so on. In Table 4, for example, replace “34.7 ± 1.7” with “35 ± 2”; “22.57 ± 0.28” with “22.6 ± 0.3”, and so on. Please correct anywhere.

There are some grammatical and spelling errors.

Specific comments

Line 143: specify the acronym TAN

Line 174-176. Authors should specify at which time the reported data are correlated: 28 day?

Line 177-178. Authors report data already showed in Table 3. I suggest that any statistically significant differences between the groups be highlighted in the text.

Line 188-193. As for lines 177-178.

Table 4. 1) What do the apex letters mean in the first row? 2) Notes are missing; 3) Data require a statistical analysis.

Table 5. Missing caption

Lines 259-230. Are there some evidences in literature that BSF can metabolize antinutritional metabolites? Authors can not affirm that based on the obtained results.

Author Response

We greatly acknowledge the time and effort of the reviewers as well as the editor. This greatly helped us to revise and improve the manuscript. We revised the manuscript as suggested. All changes in tracked changes mode can be found in the attached file of the manuscript. Our response and the line numbers below refer to this tracked changes version of our manuscript.

Reviewer: 3

The manuscript entitled “Influence of substrates on the quality of Hermetia meal for fish 2 meal substitution in Nile tilapia Oreochromis niloticus” present an interesting study but it cannot be accepted in the present form. A thorough review of the work is necessary to make it publishable in such a prestigious journal. The major criticality is linked to the experimental design. In addition to the tested substrates, it is necessary to use a standard substrate for HI breeding to obtain a control group to compare the obtained results. Generally, the control substrate is the one used by the company that provides HI. The lack of a control group makes the work unacceptable.

We completely agree on the need of a control feed and the necessary standardization. Still, control feed should not be a manufacturer based formulations, which is only available via the respective manufacturer in the respective country. Currently, producer use different substrates and do not provide a standardized substrate on a larger scale. We used chicken feed as a control substrate as availability is ensured on a larger scale. This allows comparability to several other studies [17, 18, 72] that suggested chicken feed as a control. Also, all our substrates are characterized by gross composition which furthermore supports comparison between studies. We believe our chicken feed group is thus a valid control group, also reflected by a good growth performance.

Other suggestions

I suggest adding in the introduction the part concerning the description of substrates that have been tested in the literature for the growth of HI, on the basis of which probably the authors have chosen the substrates to be tested in this research.

In the introduction wee highlight that the substrates chosen are plant by-products that are frequently used in fish meal replacement. This is the first study using these by-products as substrate to grow HI. We believe that the introduction is adeequate. We added information on the choice of substrate in the M&M as suggested by reviewer 2.

Pay attention to the significant figures. The general rule is that experimental uncertainty (Standard Deviation) should be rounded to one significant digit. In Tables and along the text, authors reported too many digits. For example, in Table 3 replace “176.07 ± 34.0” with “176 ± 34”; “20.75 ± 3.3” with “20.8 ± 3”, “89.64 ± 36.5” with “90 ± 36” and so on. In Table 4, for example, replace “34.7 ± 1.7” with “35 ± 2”; “22.57 ± 0.28” with “22.6 ± 0.3”, and so on. Please correct anywhere.

All table were changed correspondingly

There are some grammatical and spelling errors.

Specific comments

Line 143: specify the acronym TAN

Done

Line 174-176. Authors should specify at which time the reported data are correlated: 28 day?

Changed: Correlated with the mortalities, growth performance after 28 d was best in C and S groups with 176 ± 34 mg and 119 ± 11 mg (Fig. 2), respectively.

Line 177-178. Authors report data already showed in Table 3. I suggest that any statistically significant differences between the groups be highlighted in the text.

Line 188-193. As for lines 177-178.

Table 4. 1) What do the apex letters mean in the first row? 2) Notes are missing; 3) Data require a statistical analysis.

Letters were deleted, notes were added. Statistical differences were added.

Table 5. Missing caption

Added

Lines 259-230. Are there some evidences in literature that BSF can metabolize antinutritional metabolites? Authors can not affirm that based on the obtained results.

No, to our knowledge there are no data available. We therefore reformulated the sentence:

It is interesting that reduced growth of larvae but not of the respective fish group was observed here, suggesting that the glycoalkaloides do not bioaccumulate in the larvae. One may even speculate that these are metabolized or not incorporated.

Reviewer 4 Report

A well written paper in general - some very minor English edits only (see below). Does not attempt to consider amino acid profiles or other aspects of this debate but as an article about substrates, it does offer some valuable insights and interesting findings, particularly in relation to protein levels in meals and their effect on larval development versus fish growth. Would be good to see results when CF is low to start with.

Some minor edit comments

L18 "As a through" - don't understand what this means?

L172 - "P" has been left out

L208 - remove comma after "both"

L238 - proof not prove

L239 - I think this sentence needs re-phrasing; it doesn't make sense as written

There are mixed reference styles with the authors being referred to mid-sentence? e.g., 231, 237, 247, 248. I would remove author names and restructure the sentence?

L263 - 'of utmost importance'...'to assure consistency in the quality of the respective meal'

L286 - spelling "made"

Author Response

We greatly acknowledge the time and effort of the reviewers as well as the editor. This greatly helped us to revise and improve the manuscript. We revised the manuscript as suggested. All changes in tracked changes mode can be found in the attached file of the manuscript. Our response and the line numbers below refer to this tracked changes version of our manuscript.

A well written paper in general - some very minor English edits only (see below). Does not attempt to consider amino acid profiles or other aspects of this debate but as an article about substrates, it does offer some valuable insights and interesting findings, particularly in relation to protein levels in meals and their effect on larval development versus fish growth. Would be good to see results when CF is low to start with.

Some minor edit comments

L18 "As a through" - don't understand what this means?

We could not find this formulation in the text. L18 – as a conclusion

L172 - "P" has been left out

No, the following sentence states: Highest mortalities were observed in larvae grown on potato protein with 47% mortality.

L208 - remove comma after "both"

done

L238 - proof not prove

done

L239 - I think this sentence needs re-phrasing; it doesn't make sense as written

Changed to: Also, to produce meals with high protein levels protein rich substrates are needed.

There are mixed reference styles with the authors being referred to mid-sentence? e.g., 231, 237, 247, 248. I would remove author names and restructure the sentence?

The reference style has been corrected to xy et al. [xy] throughout the text

L263 - 'of utmost importance'...'to assure consistency in the quality of the respective meal'

Sentence has been rearranged

L286 - spelling "made"

done

Round 2

Reviewer 3 Report

The manuscript entitled “Influence of substrates on the quality of Hermetia meal for fish 2 meal substitution in Nile tilapia Oreochromis niloticus” present an interesting study and, after minor revisions, it deserves to be published.

Specific comments

Line 176-178. Authors did not answer to the following comment: “Authors report data already showed in Table 3. I suggest that any statistically significant differences between the groups be highlighted in the text.”

Line 207-212. Authors did not answer to the following comment: “As for lines 177-178”.

Line 237. Please add the following reference: “Physiological responses of Siberian sturgeon (Acipenser baerii) juveniles fed on full-fat insect-based diet in an aquaponic system. Matteo Zarantoniello, Basilio Randazzo, Valentina Nozzi, Cristina Truzzi, Elisabetta Giorgini, Gloriana Cardinaletti, Lorenzo Freddi, Stefano Ratti, Federico Girolametti, Andrea Osimani, Valentina Notarstefano, Vesna Milanović, Paola Riolo, Nunzio Isidoro, Francesca Tulli, Giorgia Gioacchini, Ike Olivotto. Scientific Reports, 2021, 11:1057. 10.1038/s41598-020-80379-x

Line 284-285. Replace “brown algae” with “brown algae or microalgae”. Add the following reference: Matteo Zarantoniello, Basilio Randazzo, Giorgia Gioacchini, Cristina Truzzi, Elisabetta Giorgini, Paola Riolo, Giorgia Gioia, Cristiano Bertolucci, Andrea Osimani, Gloriana Cardinaletti, Tyrone Lucon-Xiccato, Vesna Milanović, Anna Annibaldi, Francesca Tulli, Valentina Notarstefano, Sara Ruschioni, Francesca Clementi & Ike Olivotto. Zebrafish (Danio rerio) physiological and behavioural responses to insect-based diets: a multidisciplinary approach. Scientific Reports, (2020), 10(1). 10648 10.1038/s41598-020-67740-w  

Author Response

The manuscript entitled “Influence of substrates on the quality of Hermetia meal for fish 2 meal substitution in Nile tilapia Oreochromis niloticus” present an interesting study and, after minor revisions, it deserves to be published.

Specific comments

Line 176-178. Authors did not answer to the following comment: “Authors report data already showed in Table 3. I suggest that any statistically significant differences between the groups be highlighted in the text.”

We performed a multiple comparison, identifying different levels of significance between groups. This is identified in the figure using lettres for significance. In thee text such an illustration would be very confusing and we decided to leave this aspect to the figure.

Line 207-212. Authors did not answer to the following comment: “As for lines 177-178”.

Again, significance levels after multiple comparision is identified in the table. We believe that a presentation of these levels would be very confusing to the reader and decided to leave it to the table.

Line 237. Please add the following reference: “Physiological responses of Siberian sturgeon (Acipenser baerii) juveniles fed on full-fat insect-based diet in an aquaponic system. Matteo Zarantoniello, Basilio Randazzo, Valentina Nozzi, Cristina Truzzi, Elisabetta Giorgini, Gloriana Cardinaletti, Lorenzo Freddi, Stefano Ratti, Federico Girolametti, Andrea Osimani, Valentina Notarstefano, Vesna Milanović, Paola Riolo, Nunzio Isidoro, Francesca Tulli, Giorgia Gioacchini, Ike Olivotto. Scientific Reports, 2021, 11:1057. 10.1038/s41598-020-80379-x

Line 284-285. Replace “brown algae” with “brown algae or microalgae”. Add the following reference: Matteo Zarantoniello, Basilio Randazzo, Giorgia Gioacchini, Cristina Truzzi, Elisabetta Giorgini, Paola Riolo, Giorgia Gioia, Cristiano Bertolucci, Andrea Osimani, Gloriana Cardinaletti, Tyrone Lucon-Xiccato, Vesna Milanović, Anna Annibaldi, Francesca Tulli, Valentina Notarstefano, Sara Ruschioni, Francesca Clementi & Ike Olivotto. Zebrafish (Danio rerio) physiological and behavioural responses to insect-based diets: a multidisciplinary approach. Scientific Reports, (2020), 10(1). 10648 10.1038/s41598-020-67740-w

Both reeferences were inculded